# Practical Application of Drive-By Monitoring Technology to Road Roughness Estimation Using Buses in Service

**DOI:** 10.3390/s23042004

**Published:** 2023-02-10

**Authors:** Kyosuke Yamamoto, Ryota Shin, Katsuki Sakuma, Masaaki Ono, Yukihiko Okada

**Affiliations:** 1Institute of Systems and Information Engineering/Center for Artificial Intelligence Research, University of Tsukuba, 1-1-1 Ten-No-Dai, Tsukuba 305-8573, Japan; 2Graduate School of Science and Technology, University of Tsukuba, 1-1-1 Tennodai, Tsukuba 305-8577, Japan; 3Graduate School of Science and Technology, University of Tsukuba, 1-1-1 Tennodai, Tsukuba 305-8573, Japan; 4Technical Service Office for Systems and Information Engineering, University of Tsukuba, 1-1-1 Tennodai, Tsukuba 305-8573, Japan; 5Institute of Systems and Information Engineering/Center for Artificial Intelligence Research, University of Tsukuba, 1-1-1 Ten-No-Dai, Tsukuba 305-8577, Japan

**Keywords:** drive-by monitoring, vehicle, system identification, GPS synchronization, field test

## Abstract

The efficiency of vehicles and travel comfort are maintained by the effective management of road pavement conditions. Pavement conditions can be inspected at a low cost by drive-by monitoring technology. Drive-by monitoring technology is a method of collecting data from sensors installed on a running vehicle. This technique enables quick and low-cost inspections. However, most existing technologies assume that the vehicle runs at a constant speed. Therefore, this study devises a theoretical framework that estimates road unevenness without prior information about the vehicle’s mechanical parameters even when the running speed changes. This paper also shows the required function of sensors for this scheme. The required ability is to collect the three-axis acceleration vibration and position data simultaneously. A field experiment was performed to examine the applicability of sensors with both functions to the proposed methods. Each sensor was installed on a bus in service in this field experiment. The vehicle’s natural frequency estimated from the measured data ranges from 1 to 2 Hz, but the natural frequency estimated by the proposed method is 0.71 Hz. However, the estimated road unevenness does not change significantly with changes in the vehicle’s estimated parameters. The results found that the accuracy of road unevenness estimation seems to be acceptable with the conventional method and the new method. Future work will include improving the algorithm and accuracy verification of the schemes.

## 1. Introduction

Road traffic networks have expanded rapidly since motorization in the 1960s. Today, paved road networks are indispensable for people’s lives and industrial activities. However, asphalt pavement deteriorates day by day due to repeated traffic loads. Therefore, road inspections are frequently required. Pavement inspection usually consists of an appearance inspection and measurement of the spatial distribution of road unevenness. While the appearance inspection can be performed to check cracks and ruts on the pavement visually, the measurement of spatial distribution requires obtaining the absolute displacement of the pavement from the standard elevation. The latter creates a heavy workload.

The measurement of road unevenness can be classified into several types. The most reliable and accurate method is leveling. However, it is challenging to set leveling measurement points with sufficient resolution to evaluate the comfort and safety of running vehicles. Therefore, road unevenness is generally measured using a road profiler or a laser displacement measurement vehicle. The former is highly accurate, but it cannot measure rough roads as its measurement range is small. In addition, as a road profiler scans road surfaces slowly, the required measurement time is relatively long [1]. The latter has no such problem but is extremely expensive [2]. In developing countries such as Africa, large-scale road construction is underway, and a more efficient road evaluation method is required. Note that the International Roughness Index (IRI) is used for investment decisions at the World Bank [3].

Therefore, a road unevenness evaluation method using drive-by monitoring [4,5,6] has been proposed. McGetrick et al. [7] estimate the dynamic axle force from the vehicle vibration and assess the road unevenness. Zhao et al. [8] propose a technique to estimate the IRI from vehicle vibration measured by a smartphone. Zhao et al. [9] also use the Kalman filter to estimate the IRI and the road unevenness with high accuracy in field tests. Xue et al. [10] propose a method that does not require the calibration of vehicle parameters. It has been confirmed that Xue’s method is robust to changes in vehicle speed. He et al. [11] estimate bridge vibration and road unevenness from vehicle vibration using a Kalman filter. Yang et al. [1] estimate bridge vibration components and extract road unevenness from vehicle vibration. The road unevenness is evaluated by removing the bridge deflection from the unknown input of the vehicle system, calculated using the Kalman filter. Road surface information can be acquired more frequently and more economically by these drive-by monitoring technologies [1,6].

The coefficients of the IRI acceleration regression model vary greatly with vehicle speed [12]. Therefore, some methods have been proposed to deal with vehicle speed fluctuations. These schemes can be classified into (1) correction coefficients, (2) regression parameter variations, and (3) high-pass filters [2]. However, as the vehicle usually stops, there is a limitation in mitigating the effect of vehicle speed using the correction factor. Wang et al. [13] show that different vehicle types have different magnitudes of acceleration responses to the same road profile. In addition, the coefficients of the regression model differ according to the type of vehicle [14]. Therefore, correction based on vehicle vibration is premised on accumulating multiple data.

To fully demonstrate the potential advantages of drive-by monitoring technologies, such as high frequency and economic performance, it is essential to apply this method to general vehicles. However, it is difficult and unsafe for ordinary drivers to drive their vehicles at a constant speed for road roughness estimation. The effect of speed changes is an unavoidable issue with regard to the social implementation of large-scale drive-by monitoring.

Numerical simulations have been conducted to simulate vehicle vibration changes due to speed fluctuations and to evaluate the IRI from the obtained vehicle vibration with high accuracy [13,15]. As Yu et al. [2] point out, Moghadam [15] did not investigate the responses under the condition of speed fluctuations. In addition, Wang et al. [13] have not been able to consider realistic speed fluctuations yet. Thus, one of the unsolved problems of drive-by monitoring is vehicle speed fluctuations. The proposed method by Xue et al. [10] shows high accuracy in field tests even when considering vehicle speed fluctuations. Their method uses time-domain analysis, which makes it less sensitive to changes in vehicle speed. Therefore, the problem of vehicle speed changes has not been solved but ignored in this scheme. Keenahan et al. [16] verify a similar scheme to estimate road unevenness from measured vehicle vibration and randomly assumed parameters.

Thus, this paper proposes a method for estimating road unevenness under the condition of speed changes. This study does not measure the correct value of road unevenness. This research aims to construct a sensor system that enables the proposed scheme and confirms that the proposed method can be performed practically by the developed devices with each having an accelerometer and a global navigation satellite system (GNSS) unit.

The academic contributions of this research are as follows:This study shows two methods that can estimate the input road unevenness from vehicle vibration data even without information on the vehicle’s mechanical parameters.Applying the proposed method to vehicle vibration data both of vertical and traveling directions, the road unevenness angle and curvature can be estimated.The suggested device with a vibration sensor and a GNSS device makes the proposed method practical. Position synchronization realizes this.

According to a previous study [16], the road unevenness estimated by synchronizing the positions of multiple vehicles is also equal to the correct values.

## 2. Sensor System

### 2.1. Concept of Sensor

A GNSS receiver and a three-axis accelerometer were installed on four traveling buses in service. The GNSS unit used in this study receives satellite signals at 1 Hz. The satellite signals contain position information and accurate time within ±100 ns. This accurate time was used to synchronize the clocks of all accelerometers in the different buses. The accuracy of this GNSS-based time synchronization can be realized by the pulse per second (PPS) signals of the GNSS receivers. The position information is acquired after correction in the GNSS unit. Therefore, a time lag occurs. This time lag is corrected by referring to the acquisition time of the corresponding PPS signal. The clock time, the vehicle position, and the acceleration vibration data are simultaneously recorded at 300 Hz. As the sampling rate of the position information is 1 Hz, it is converted to 300 Hz by interpolation.

### 2.2. Specification and Function

Figure 1 shows a block diagram of the Zynq chip on the ZYBO Z7-10 board (by Xilinx, Inc., San Jose, CA, USA) used for the vibration measurement system. The vibration measurement system consists of a ZYNQ7 Processing System (by Xilinx, Inc., San Jose, CA, USA) powered by an ARM processor (by ARM Ltd., Cambridge, England, UK), a reset circuit, and two field-programmable gate array (FPGA)-based MicroBlaze (MB) processors (by Xilinx, Inc., San Jose, CA, USA) named mb_iic and gps_uartlite. ADXL355 (by Analog Devices Inc., Norwood, MA, USA) is adopted as the 3-axis accelerometer module in this system; this module has 20-bit ADC resolution and ±8 G range. ADXL355 and the GNSS receiver (AE-GYSFDMAXB by TAIYO TUDEN, Tokyo, Japan) are mounted on an expansion board. The communication standard between ZYBO Z7-10 and ADXL355 is the I2C interface, while the serial interface is used between ZYBO Z7-10 and the GNSS unit. The ARM processor operates the FPGA-based MB processors and writes measured data to the MicroSD card every 3.333 ms. As the data are also acquired every 3.333 ms, the writing time of the MicroSD card fluctuates. Therefore, data acquisition is forcibly performed by interrupts, while writing to the MicroSD card is asynchronously performed through the first in, first out (FIFO) block memories. 

Figure 2 shows a block diagram of the mb_iic module. The mb_iic module is equipped with an MB processor, which controls the 3-axis acceleration sensor. The general-purpose input/output (GPIO) connected to the MB processor and the GPIO connected to the ARM processor are linked through parallel ports allowing data exchange. When a start command is sent from the ARM processor, the MB processor acquires the values of the 3-axis acceleration sensor. It then writes those values to the GPIO along with the launch command. The ARM processor then receives the startup code and data. Figure 3 also shows a block diagram of the gps_uartlite module. This processor receives serial data from the GNSS unit, edits the data, and passes the data to the ARM processor via GPIO.

A photograph of this vibration sensor system is shown in Figure 4. The accelerometer ADXL355 and the GNSS unit are installed on the extension board. This board is connected to the ZYBO Z7-10 board through PMOD connectors, as shown in this photograph. This sensor system is slightly over-specified when considering only the vehicle vibration measurement.

The saved data of the suggested sensor system consist of (1) longitude and latitude (GPRCMC: dddmm.mmmm), (2) GNSS time (coordinated universal time), (3) PPS, (4) CPU time (ms), (5) three-axis accelerations (travel, lateral, and vertical directions). The positioning information and the clock time from the GNSS device are automatically calculated and updated inside the GNSS. Therefore, the time lag between when the radio waves from the satellite reach the receiver and when the updated position/time information is acquired is calculated from the PPS time. The position information updated at 1 Hz is interpolated into the 300 Hz data. Then, the distance $x_{i}\left(t\right)$ is calculated, where i represents the vehicle number and t is the GNSS clock time. Note that t=0 is determined appropriately. As the vertical acceleration response, ${\ddot{z}}_{i}\left(t\right)$, is related to $x_{i}\left(t\right)$, it can be easily transformed into the spatial function ${\ddot{z}}_{i}\left(x\right)$.

## 3. Road Unevenness Estimation

Signal processing for estimating road unevenness from vehicle vibration can be considered an input estimation problem. The easiest way to solve this problem is the application of the frequency response function (FRF) of the monitoring vehicle. If the FRF is known, the product of the inverse FRF and measured vehicle vibration in the frequency domain becomes the road profile input. However, this method has two technical issues: (1) the FRF of the vehicle is required; (2) it is difficult to consider the effect of vehicle speed fluctuations. Thus, this paper presents a new method to estimate road unevenness components from vehicle vibration data without prior information about the vehicle’s mechanical parameters and overcome the above two issues.

First, to avoid measurement of the FRF, all possible combinations of the mechanical parameters of the vehicle are enumerated. The most likely combination of the parameters can be identified to find the combination with the least contradiction. However, the remaining technical issue is the difficulty of how to find an indicator for evaluating inconsistency among measured data, assumed parameters, and equation of motion. For this issue, Xue’s method [10] and Keenahan’s method [16] are applicable.

Second, to consider vehicle speed fluctuations, it is necessary to manage the road profile components affected by the acceleration and speed of the traveling vehicle. This paper adopts the time derivatives of road unevenness spatial functions. 

### 3.1. Vehicle System

The vehicle system can be modeled as a simple mass-spring-dashpot system [4,5,6], shown in Figure 5. The vehicle mass, m, includes the total mass of the vehicle body, tires, suspension, engine, fuel, other mechanical parts, and passengers. The spring and damping coefficients, k and c, respectively, mainly represent the performance of the vehicular suspension.

Letting $z_{i}\left(t\right)$ and $r_{i}\left(t\right)$ be the vehicle vibration displacement and the road profile of the i-th vehicle, respectively, the equation of motion of the i-th vehicle is expressed as
(1)$$m{\ddot{z}}_{i}\left(t\right)+c{\dot{z}}_{i}\left(t\right)+kz_{i}\left(t\right)=c{\dot{r}}_{i}\left(t\right)+kr_{i}\left(t\right)$$
where $\dot{\left(\;\right)}$ and $\ddot{\left(\;\right)}$ represent the first-order and second-order derivatives with respect to time, t. The road profile, $r_{i}\left(t\right)$, is originally a spatial function but is transformed to a time function by using the vehicle position, $x_{i}\left(t\right)$.
(2)$$r_{i}\left(t\right)=R\left(x_{i}\left(t\right)\right)$$
It is noted that the road unevenness, $R\left(x\right)$, does not change if the driving route is the same, while the road profile, $r_{i}\left(t\right)$, changes due to the running speed. Fourier’s transform of Equation (1) is given by the following formula:(3)$${\hat{z}}_{i}\left(\omega \right)=\hat{h}\left(\omega \right){\hat{r}}_{i}\left(\omega \right)$$
where $\hat{h}\left(\omega \right)$ is the frequency response function (FRF) and ω represents the angular frequency.
(4)$$\hat{h}\left(\omega \right)=\frac{ic\omega +k}{-m\omega^{2}+ic\omega +k}=\frac{j\left(\frac{c}{m}\right)\omega +\left(\frac{k}{m}\right)}{-\omega^{2}+j\left(\frac{c}{m}\right)\omega +\left(\frac{k}{m}\right)}$$
where j is the imaginary unit. It is assumed that the FRF of the vehicle does not change even in repeated runs. When the vehicle’s acceleration responses, ${\ddot{z}}_{i}\left(t\right)$, are available, the road unevenness, $R\left(x\right)$, can be estimated from the following equation:(5)$${\hat{r}}_{i}\left(\omega \right)=\frac{{\hat{z}}_{i}\left(\omega \right)}{\hat{h}\left(\omega \right)}=\frac{-\omega^{2}{\hat{z}}_{i}\left(\omega \right)}{-\omega^{2}\hat{h}\left(\omega \right)}=\frac{\left({\hat{\ddot{z}}}_{i}\left(\omega \right)\right)}{-\omega^{2}\hat{h}\left(\omega \right)}$$
because ${\hat{\ddot{z}}}_{i}\left(\omega \right)=-\omega^{2}{\hat{z}}_{i}\left(\omega \right)$. The conventional method for estimating road profiles from vehicle vibrations has been based on Equation (5). The road profile, $r_{i}\left(t\right)$, can be obtained by the inverse Fourier transform of ${\hat{r}}_{i}\left(\omega \right)$.

As the process of Equation (5) can be considered to include numerical integration, the robustness of this process is low. If the FRF is directly applied to the road profile estimation instead of $-\omega^{2}\hat{h}\left(\omega \right)$ to avoid numerical integration, the obtained signal becomes ${\ddot{r}}_{i}\left(t\right)$:(6)$${\hat{\ddot{r}}}_{i}\left(\omega \right)=\frac{{\hat{\ddot{z}}}_{i}\left(\omega \right)}{\hat{h}\left(\omega \right)}$$

This process can be robust. However, the road profile acceleration, ${\ddot{r}}_{i}\left(t\right)$, derived from the inverse Fourier transform of ${\hat{\ddot{r}}}_{i}\left(\omega \right)$ includes the effect of running speed $v_{i}\left(t\right)={\dot{x}}_{i}\left(t\right)$ and acceleration/deceleration $a_{i}\left(t\right)={\ddot{x}}_{i}\left(t\right)$ as the following formula.
(7)$${\ddot{r}}_{i}\left(t\right)=\frac{d^{2}}{dt^{2}}R\left(x_{i}\left(t\right)\right)=a_{i}\left(t\right)R^{\prime }\left(x_{i}\left(t\right)\right)+v_{i}{\left(t\right)}^{2}R^{\prime \prime }\left(x_{i}\left(t\right)\right)$$
where ${\left(\;\right)}^{\prime }$ and ${\left(\;\right)}^{\prime \prime }$ represent the first-order and second-order derivatives with respect to position x. $R^{\prime }\left(x\right)$ and $R^{\prime \prime }\left(x\right)$ are common like $R\left(x\right)$. However, the obtained input ${\ddot{r}}_{i}\left(t\right)$ includes $a_{i}\left(t\right)$ and $v_{i}\left(t\right)$, which change in every run. As this variation makes the evaluation of ${\ddot{r}}_{i}\left(t\right)$ difficult, road unevenness, $R\left(x\right)$, has been mainly used in existing studies [10,16,17,18].

### 3.2. Traditional Method

The measurement of vehicle parameters is usually complicated. Many existing studies assume that the vehicle parameters are known [2,7,11] or calibrated from vehicle vibrations [8,9]. However, Xue et al. [10] and Keenahan et al. [16] assume the parameters randomly first and apply a genetic algorithm (GA) to vehicle vibration data to simultaneously estimate the road profile and vehicle parameters. 

Therefore, this paper considers a method of estimating vehicle parameters $\left(\frac{c}{m}\right)$ and $\left(\frac{k}{m}\right)$ and road unevenness, $R\left(x\right)$, using only the measured vehicle vibration, ${\ddot{z}}_{i}\left(t\right)$. It is assumed that the vehicle travels the same route many times. 

The algorithm is shown in Figure 6. First, the vehicle acceleration vibration, ${\ddot{z}}_{i}\left(t\right)$, is measured and the parameters $\left(\frac{c}{m}\right)$ and $\left(\frac{k}{m}\right)$ are randomly assumed. By substituting these data and properties into Equation (5), the road unevenness, $R_{i}\left(x\right)$, can be obtained. Let $\bar{R}\left(x\right)$ be the mean of $R_{i}\left(x\right)$:(8)$$\bar{R}\left(x\right)=\frac{1}{n}\overset{n}{\underset{i=1}{\sum }}R_{i}\left(x\right)$$
where n is the number of repeated runs. The likelihood of the vehicle parameters $\left(\frac{c}{m}\right)$ and $\left(\frac{k}{m}\right)$ can be evaluated by the following formula:(9)$$J_{1}=\overset{n}{\underset{i=1}{\sum }}\overset{N}{\underset{k=1}{\sum }}{\left(R_{i}\left(x_{k}\right)-\bar{R}\left(x_{k}\right)\right)}^{2}$$
where $x_{k}$ denotes the position of the k-th representative point and N is the number of representative points. The most likely vehicle parameters $\left(\frac{c}{m}\right)$ and $\left(\frac{k}{m}\right)$ are those that minimize the matching degree of road unevenness, $J_{1}$, calculated by Equation (9), and the road unevenness obtained at that time can be taken as the estimated road unevenness, $R\left(x\right)$.

The scheme shown above is a simplified version of Xue’s drive-by process, so it is not new. However, as Xue et al. [10] estimate many vehicle parameters simultaneously, searching for the optimal solution in a vast space is necessary. That is the reason why they adopt a GA. However, this method limits the number of vehicle parameters to only two, $\left(\frac{c}{m}\right)$ and $\left(\frac{k}{m}\right)$. Therefore, the computational cost is significantly reduced and it is possible to try all patterns. Note that it is necessary to measure the position information of the vehicle as well as the vibration data to execute this method. 

### 3.3. The New Method

The novel scheme for evaluating road unevenness using $R^{\prime }\left(x\right)$ and $R^{\prime \prime }\left(x\right)$ is shown in Figure 7. This scheme is based on input estimation without numerical integration, even considering vehicle speed changes.

First, in this process, the vehicle’s vertical acceleration vibrations, ${\ddot{z}}_{i}\left(t\right)$, and acceleration of the traveling direction, $a_{i}\left(t\right)$, ($={\ddot{x}}_{i}\left(t\right)$) are measured directly by the vibration sensor on the vehicle. The vehicle’s position, $x_{i}\left(t\right)$, is also measured by the GNSS device. On the other hand, the vehicle parameters $\left(\frac{c}{m}\right)$ and $\left(\frac{k}{m}\right)$ are randomly assumed. The speed, $v_{i}\left(t\right)$, ($={\dot{x}}_{i}\left(t\right)$) can be estimated by the measured position, $x_{i}\left(t\right)$. Then, ${\ddot{r}}_{i}\left(t\right)$ is obtained from Equation (6). The measured time history data can be synchronized spatially. For example, ${\ddot{r}}_{i}\left(t\right)$ can be rewritten in ${\ddot{r}}_{i}\left(x\right)$ using $x_{i}\left(t\right)$. Similarly, $v_{i}\left(t\right)$ and $a_{i}\left(t\right)$ can become $v_{i}\left(x\right)$ and $a_{i}\left(x\right)$. Thus, the relationship between the available time history data (${\ddot{r}}_{i}\left(x\right)$, $v_{i}\left(x\right)$, and $a_{i}\left(x\right)$) and the road unevenness components ($R^{\prime }\left(x\right)$ and $R^{\prime \prime }\left(x\right)$) can be expressed by the following formula:(10)$$\ddot{r}\left(x\right)=A\left(x\right)\left\{\begin{array}{c} R^{\prime }\left(x\right)\\ R^{\prime \prime }\left(x\right) \end{array}\right\}$$
where
(11)$$\ddot{r}\left(x\right)=\left\{\begin{array}{c} {\ddot{r}}_{1}\left(x\right)\\ \vdots \\ {\ddot{r}}_{n}\left(x\right) \end{array}\right\}$$
(12)$$A\left(x\right)=\left[\begin{array}{cc} a_{1}\left(x\right) & v_{1}{\left(x\right)}^{2}\\ \vdots & \vdots \\ a_{n}\left(x\right) & v_{n}{\left(x\right)}^{2} \end{array}\right]$$

As the number of repeated runs, n, is usually more than two, the road unevenness components $R^{\prime }\left(x\right)$ and $R^{\prime \prime }\left(x\right)$ can be estimated by the following formula:(13)$$\left\{\begin{array}{c} R^{\prime }\left(x\right)\\ R^{\prime \prime }\left(x\right) \end{array}\right\}=A{\left(x\right)}^{+}\ddot{r}\left(x\right)$$
where $A^{+}$ denotes the pseudo-inverse matrix of A. The following formula can evaluate the estimation error:(14)$$\varepsilon \left(x\right)=\ddot{r}\left(x\right)-A\left(x\right)A{\left(x\right)}^{+}\ddot{r}\left(x\right)$$

Thus, the sum of the squared error can be expressed as
(15)$$J_{2}=\overset{N}{\underset{k=1}{\sum }}\varepsilon {\left(x_{k}\right)}^{T}\varepsilon \left(x_{k}\right)$$

$J_{2}$ can be used as the objective function to update the randomly assumed vehicle parameters $\left(\frac{c}{m}\right)$ and $\left(\frac{k}{m}\right)$. The most likely vehicle parameters $\left(\frac{c}{m}\right)$ and $\left(\frac{k}{m}\right)$ are those that minimize the matching degree of road unevenness, $J_{2}$, calculated by Equation (15), and the road unevenness obtained at that time can be taken as the estimated road unevenness components $R^{\prime }\left(x\right)$ and $R^{\prime \prime }\left(x\right)$. 

The originality of this method is to be able to consider vehicle speed changes. Conventional methods cannot consider speed changes (acceleration/deceleration) and assume a constant running speed. However, this condition limits the data that can be analyzed. This study is one step toward extending the applicability of drive-by monitoring.

Both the conventional method and the proposed method shown in this study require measured vibration data not only in the vertical direction, ${\ddot{z}}_{i}\left(t\right)$, but also in the traveling direction, $a_{i}\left(t\right)$. In addition, a GNSS device on the sensor system is used to measure the running speed, $v_{i}\left(t\right)$, of the vehicle, as well as the position, $x_{i}\left(t\right)$. Thus, sensors used for this drive-by monitoring must provide the above functions.

## 4. Field Test

### 4.1. Experimental Setting

A field test using four buses, each with one sensor shown in Figure 4, was conducted to confirm that the schemes shown in Figure 6 (the traditional method) and Figure 7 (the new method) can be practiced. In this experiment, the buses traveled between two bus terminals near Kigali, Rwanda. The fundamental theory of the introduced methods assumes that one identical vehicle would repeatedly run. However, in this field test, four buses of the same type ran once each. Then, the four vehicle vibration data obtained were regarded as one bus repeatedly traveling four times.

Figure 8 shows a photograph of one of the target buses. The sensor system was fixed to the floor inside the bus. The sensor system consists of an accelerometer and a GNSS unit. While the vertical acceleration vibration, ${\ddot{z}}_{i}\left(t\right)$, and the acceleration/deceleration, $a_{i}\left(t\right)$, are obtained from the accelerometer, the vehicle position, $x_{i}\left(t\right)$, and velocity, $v_{i}\left(t\right)$, are calculated from the latitude and longitude information obtained from the GNSS unit. The routes of the buses are also shown in Figure 9. The distance from the start to the endpoint is about 21.6 km. The jagged red lines on the maps indicate the vertical acceleration responses, ${\ddot{z}}_{i}\left(t\right)$, of the buses. The route of this field experiment is paved highway near Kigali city, Rwanda. On the day of the experiment, it was sunny and the temperature was 27 °C.

### 4.2. Measured Data

The vibrations of the travel, lateral, and vertical directions measured by the vibration sensor (${\ddot{x}}_{i}\left(t\right)$, ${\ddot{y}}_{i}\left(t\right)$, and ${\ddot{z}}_{i}\left(t\right)$, respectively) are shown in Figure 10. The sampling rate of the vibration sensors is 300 Hz. The GNSS unit synchronizes the sensor clocks, as shown in this figure. The vertical accelerations, ${\ddot{z}}_{i}\left(t\right)$, shown in Figure 10c include the gravity acceleration, 1 G. While a vehicle is running, the amplitude becomes large. According to these figures, the timing at which the amplitude increases differs depending on the bus because the buses depart and arrive at different times. Figure 11 shows the x–t curve of each bus. In this figure, x = 0 indicates the start bus terminal and x = 21.6 km indicates the end bus terminal.

The Fourier’s power spectra of the obtained vibration data are shown in Figure 12. The power of the vertical acceleration responses, ${\hat{\ddot{z}}}_{i}\left(\omega \right)$, can indicate vehicle natural frequencies but the tendency of each power spectrum is very different from the others. This means that the vertical responses are affected significantly by the difference in running speed fluctuations. The first predominant frequencies of the buses (i = 1 to 4) are about 1.44, 1.52, 1.61, and 1.63 Hz, respectively. The first and third buses had close modes around the first predominant modes: 1.87 and 1.96 Hz, respectively. It can be expected that the natural frequency of this type of bus ranges from 1 to 2 Hz. However, the buses showed higher predominant frequencies around 25 and 30 Hz. There is a possibility that the buses used in this test also have higher-order vibration modes around these frequencies. 

According to the GNSS data, the measured vibration data (${\ddot{x}}_{i}\left(t\right)$, ${\ddot{y}}_{i}\left(t\right)$, and ${\ddot{z}}_{i}\left(t\right)$) can be transformed into the form of spatial functions using vehicle position data, $x_{i}\left(t\right)$. The sampling rate of the GNSS devices for obtaining $x_{i}\left(t\right)$ is 1 Hz. The spatially synchronized vibrations (${\ddot{x}}_{i}\left(x\right)$, ${\ddot{y}}_{i}\left(x\right)$, and ${\ddot{z}}_{i}\left(x\right)$) are shown in Figure 13. According to existing studies [6], the vehicle vibration data become similar when spatially synchronized. However, the obtained vibration data in this experiment show different tendencies even after spatial synchronization. The different timing of the travel accelerations and decelerations may significantly affect the vehicle vibration tendencies. According to Figure 13a, as there is a certain tendency in the position where the travel accelerations, $a_{i}\left(x\right)={\ddot{x}}_{i}\left(t\right)$, become large, it is considered that the influence of the traffic environment (signals and legal limited speed) is almost the same. However, their amplitude changes because the strength of the vehicular accelerator and brake varies depending on the driver. In addition, the effect of vertical vibration, ${\ddot{z}}_{i}\left(x\right)$, can be confirmed in the traveling direction, ${\ddot{x}}_{i}\left(x\right)$, and the lateral direction, ${\ddot{y}}_{i}\left(x\right)$.

### 4.3. Results and Discussion

The assumed values for the vehicle parameters $\left(\frac{c}{m}\right)$ and $\left(\frac{k}{m}\right)$ are shown in Table 1. The natural frequency corresponding to each value of
$\left(\frac{k}{m}\right)$ is also shown. These assumed parameters are decided according to the power spectra of the vehicle responses shown in Figure 12.

These assumed parameters and the measured vibration data are applied to the road profile estimation schemes shown in Figure 6 (the traditional method) and Figure 7 (the new method), respectively. The resulting values of
J are shown in Figure 14. According to these figures, the optimized parameters differ between the traditional and new methods. As shown in Figure 14a, the traditional method has the optimal value of
$J_{1}$ calculated from Equation (9), when $\left(\frac{c}{m}\right)$ = 2 and $\left(\frac{k}{m}\right)$ = 5. On the other hand, as shown in Figure 14b, the new method minimizes $J_{2}$ calculated from Equation (15), when $\left(\frac{c}{m}\right)$ = 10 and $\left(\frac{k}{m}\right)$ = 1440.

According to Figure 14, as the mechanical parameters $\left(\frac{c}{m}\right)$ and $\left(\frac{k}{m}\right)$ optimizing $J_{1}$ and $J_{2}$ differ, it is difficult to determine the mechanical parameters $\left(\frac{c}{m}\right)$ and $\left(\frac{k}{m}\right)$ from these results. According to an existing study [17], the function shape of $J_{1}$ is convex downward. However, those of $J_{1}$ and $J_{2}$ are monotonically increasing and decreasing, respectively. Thus, it is difficult to limit the search area for $\left(\frac{c}{m}\right)$ and $\left(\frac{k}{m}\right)$.

Next, Figure 15 shows the estimated road profiles, $r_{i}\left(x\right)$, the estimated road profile accelerations, ${\ddot{r}}_{i}\left(x\right)$, and the reproduced road profile accelerations, $A\left(x\right)A{\left(x\right)}^{+}\ddot{r}\left(x\right)$, when $\left(\frac{c}{m}\right)$ = 2 and $\left(\frac{k}{m}\right)$ = 5. In the figures of $r_{i}\left(x\right)$, $\bar{R}\left(x\right)$ represents the mean of the estimated road profiles, $R_{i}\left(x\right)$, shown in Equation (8). According to the results of $R_{i}\left(x\right)$ shown in Figure 15a, because the mean value $\bar{R}\left(x\right)$ and each estimated road unevenness, $R_{i}\left(x\right)$, are similar, the estimation accuracy can be expected to be relatively high. Note that the correct value is not available in this study, and the estimation accuracy is not appropriately evaluated. On the other hand, the estimated road profile accelerations, $\ddot{R_{i}}\left(x\right)$, shown in Figure 15b are different from each other. This difference can be caused by the running speed fluctuations. Figure 15c shows the reproduced road profile accelerations, $A\left(x\right)A{\left(x\right)}^{+}\ddot{r}\left(x\right)$. If the residual between ${\ddot{R}}_{i}$ and $AA^{+}\ddot{r}$ is smaller, the estimation accuracy of the new method can be evaluated to be better. In this case, as the adopted mechanical parameters of the vehicle are not optimal for minimizing the residual, the road profile accelerations, ${\ddot{R}}_{i}\left(x\right)$, estimated from ${\hat{\ddot{r}}}_{i}\left(\omega \right)$ and ones reproduced by $AA^{+}\ddot{r}$ do not match well. However, as the tendencies of the estimated $\ddot{R}\left(x\right)$ and the reproduced $\ddot{R}\left(x\right)$ become spatially similar, the application result of the new method to the road unevenness evaluation can be reliable. The components $A^{+}\ddot{r}$ are considered the common road unevenness components $R^{\prime }\left(x\right)$ and $R^{\prime \prime }\left(x\right)$. Those when $\left(\frac{c}{m}\right)$ = 2 and $\left(\frac{k}{m}\right)$ = 5 are shown in Figure 16. From this figure, the road unevenness curvature, $R^{\prime \prime }\left(x\right)$, is much smaller than the road unevenness slope, $R^{\prime }\left(x\right)$.

To find more likely parameters, the objective function is redefined with the following formula: (16)$$J=J_{1}^{2}J_{2}=\overset{n}{\underset{i=1}{\sum }}\overset{N}{\underset{k=1}{\sum }}{\left(R_{i}\left(x_{k}\right)-\bar{R}\left(x_{k}\right)\right)}^{4}\times \overset{N}{\underset{k=1}{\sum }}\varepsilon {\left(x_{k}\right)}^{T}\varepsilon \left(x_{k}\right)$$

Because the residual between the estimated $R\left(x\right)$ and the mean $\bar{R}\left(x\right)$ has only a slight variation, the error raised to the fourth power is used for this term instead of the squared error. The redefined J is shown in Figure 17. The optimal parameters change to $\left(\frac{c}{m}\right)$ = 10 and $\left(\frac{k}{m}\right)$ = 20. The estimated vehicle frequency is 0.71 Hz, which is still different from the value expected from the power spectra shown in Figure 12. However, the value becomes relatively closer to its expected range. The estimated road profiles and roughness components when $\left(\frac{c}{m}\right)$ = 10 and $\left(\frac{k}{m}\right)$ = 20 are shown in Figure 18 and Figure 19, respectively.

When $\left(\frac{c}{m}\right)$ = 10 and $\left(\frac{k}{m}\right)$ = 20, the estimated road unevenness functions, $R\left(x\right)$, as shown in Figure 18a, are similar to each other. From this result, there is a probability that the accuracy of the vehicle parameters may not affect the estimation accuracy of the road roughness. The estimated road unevenness accelerations, $\ddot{R}\left(x\right)$, and the reproduced road unevenness acceleration, $AA^{+}\ddot{r}$, are shown in Figure 18b,c, respectively. The estimated road unevenness angle, $R^{\prime }\left(x\right)$, and curvature, $R^{\prime \prime }\left(x\right)$, when $\left(\frac{c}{m}\right)$ = 10 and $\left(\frac{k}{m}\right)$ = 20 are also shown in Figure 19. Many of the extremely large values included in $\ddot{R}\left(x\right)$ have been erased. However, the reproducing accuracy is not high because the parameters are not optimal.

The methods presented in this study use the inverse FRF to estimate vehicle inputs, while Xue et al. [10] apply the Kalman Filter and Keenahan et al. [16] apply the Newmark-beta method. The first scheme for estimating road unevenness, $R\left(x\right)$, seems to work well. According to a previous study [16], when the estimated road unevenness match, they are close to the correct value. Therefore, it can be said that the estimated result is reliable. On the other hand, as the new suggested scheme can estimate the road unevenness slope, $R^{\prime }\left(x\right)$, and curvature, $R^{\prime \prime }\left(x\right)$, the speed fluctuations of the monitoring vehicle can be considered in the process of the vehicle input estimation. However, its accuracy evaluation is also future work.

According to the obtained results, the presented schemes can be practical by the developed sensor systems. The contribution of this study is to show that the suggested vibration sensors each with a GNSS unit make the proposed methods practical. Spatial synchronization realizes these schemes.

## 5. Conclusions

This study shows two methods that can estimate the input road unevenness even without information on the vehicle’s mechanical parameters when the vehicle’s vibration and position data are obtained. The first shown method is traditional and estimates the road unevenness directly. However, as the obtained acceleration data are converted to estimated displacement data, there is a concern about the instability caused by numerical integration. Therefore, the second method uses both vertical and traveling direction vibration data to estimate the road unevenness angle and curvature. A new sensor system was designed in this study for the implementation of the above two input estimation algorithms.

Furthermore, the designed sensors were installed on four buses in commercial operation. The road unevenness and its components were estimated. In this experiment, as the correct data of the road unevenness for the travel distance of 21.6 km was not obtained, it was not possible to verify the accuracy. However, this field experiment has confirmed that the sensor system and the two road estimation schemes are practicable. Future issues include verifying the accuracy of this scheme and upgrading the algorithms by increasing the number of sensors installed in each vehicle.

In addition, if the input estimation accuracy by the drive-by monitoring technology is improved, it is expected that the bridge vibration component can be extracted [18,19,20]. However, the accuracy of the presented schemes is still insufficient for bridge inspections, because the amplitude of bridge vibration is usually smaller than road unevenness [21,22]. It is necessary to improve both the sensor system and the estimation algorithms shown in this paper to inspect bridges as well as road pavements. Therefore, one of the future works includes adding measurement items such as the installation angle of the sensor.

This study is the first case of a drive-by scheme considering travel speeds and acceleration. The results of this research provide a practical scheme for expanding the application range of drive-by monitoring from specialized vehicles to ordinary vehicles. In other words, drive-by monitoring is now practicable not only in a lab experiment where a carefully calibrated vehicle runs at constant speed but also in an environment where the drivers are driving as usual.

## Figures and Tables

**Figure 1 sensors-23-02004-f001:**
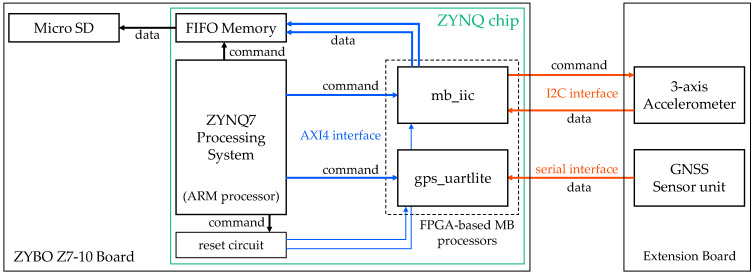
Block diagram of the Zynq chip adopted in the vibration measurement system; this sensor has two FPGA-based MB processors to obtain vibration and position data. MB processors are controlled by the ARM processor on the ZYNQ7 Processing System.

**Figure 2 sensors-23-02004-f002:**
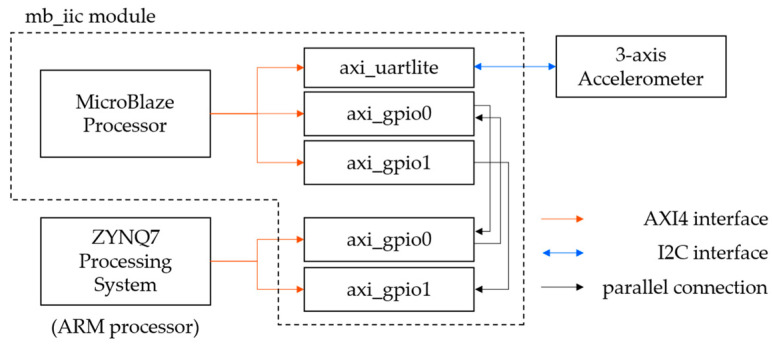
Block diagram of the mb_iic module.

**Figure 3 sensors-23-02004-f003:**
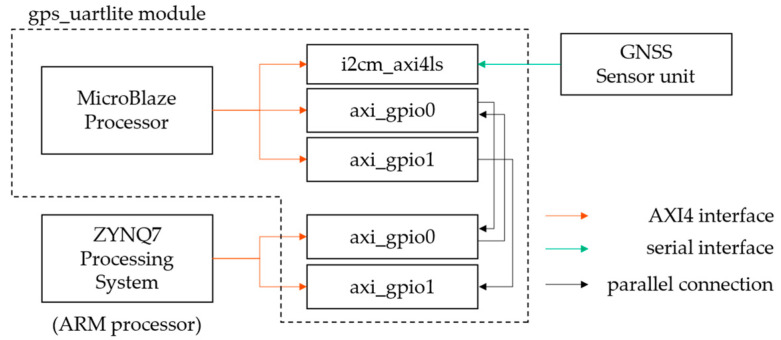
Block diagram of the gps_uartlite module.

**Figure 4 sensors-23-02004-f004:**
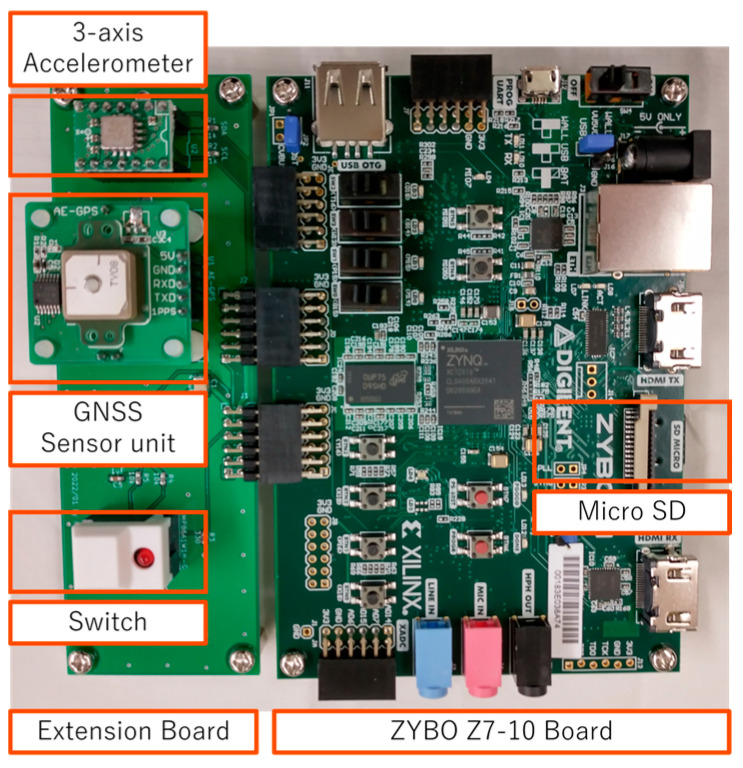
Photograph of the developed sensor in this study.

**Figure 5 sensors-23-02004-f005:**
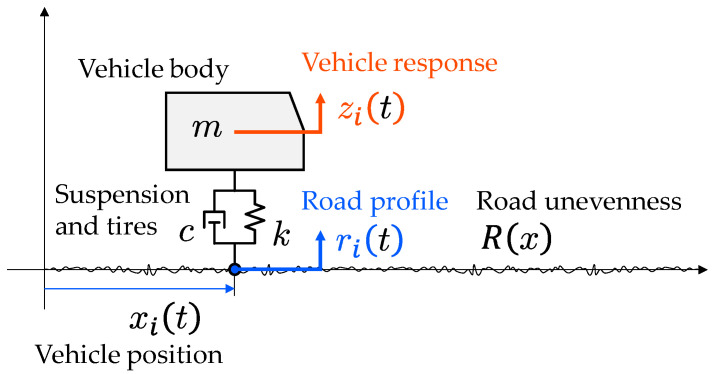
The vehicle system is modeled as a single degree of freedom (SDOF) system with mass, m, damping, c, and stiffness, k. The input of the vehicle system is only the road profile, $r_{i}\left(t\right)=R\left(x_{i}\left(t\right)\right)$, while the output is the vertical acceleration response, $z_{i}\left(t\right)$.

**Figure 6 sensors-23-02004-f006:**
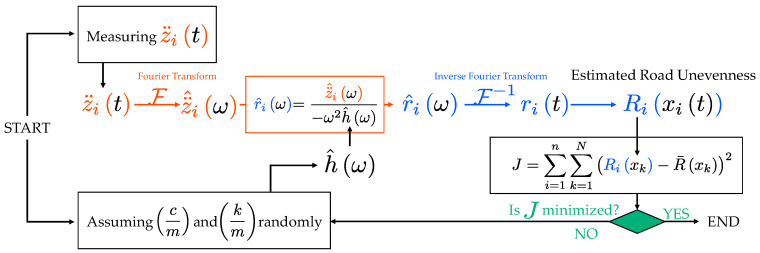
Flowchart of the traditional method of estimating road unevenness, $R\left(x\right)$, without prior information about the vehicle parameters $\left(\frac{c}{m}\right)$ and $\left(\frac{k}{m}\right)$.

**Figure 7 sensors-23-02004-f007:**
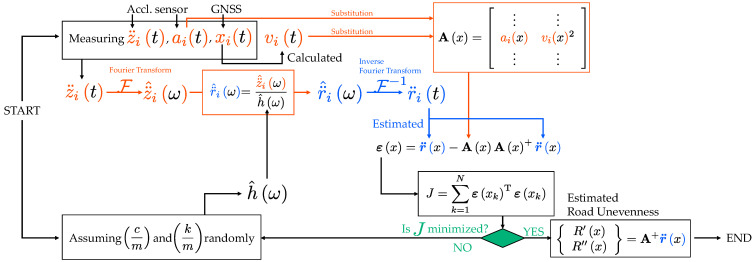
Flowchart of the proposed method of estimating road unevenness components $R^{\prime }\left(x\right)$ and $R^{\prime \prime }\left(x\right)$ without prior information about the vehicle parameters $\left(\frac{c}{m}\right)$ and $\left(\frac{k}{m}\right)$. This process does not require the vehicle’s prior information and can also estimate inputs.

**Figure 8 sensors-23-02004-f008:**
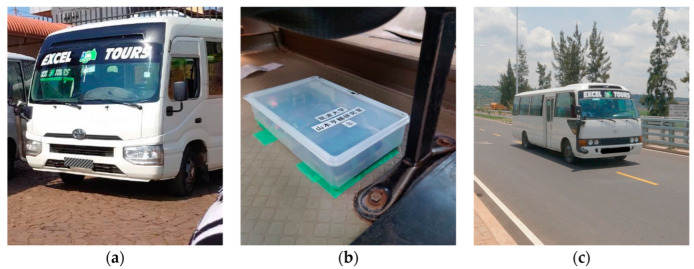
The bus for measurement: (**a**) the bus; (**b**) the sensor fixed to the floor inside the vehicle; (**c**) the bus in motion.

**Figure 9 sensors-23-02004-f009:**
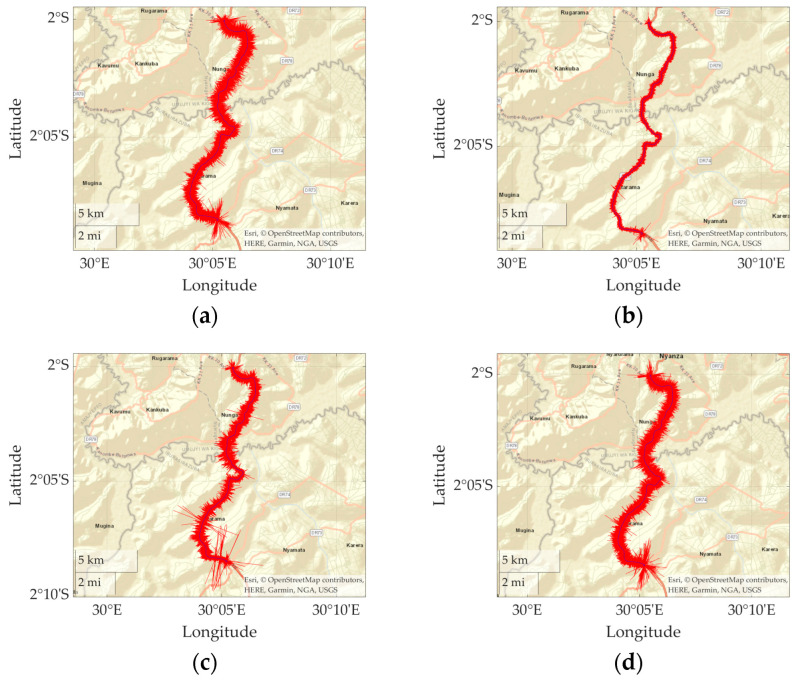
The vertical vibration on the route map: (**a**) i = 1; (**b**) i = 2; (**c**) i = 3; (**d**) i = 4.

**Figure 10 sensors-23-02004-f010:**
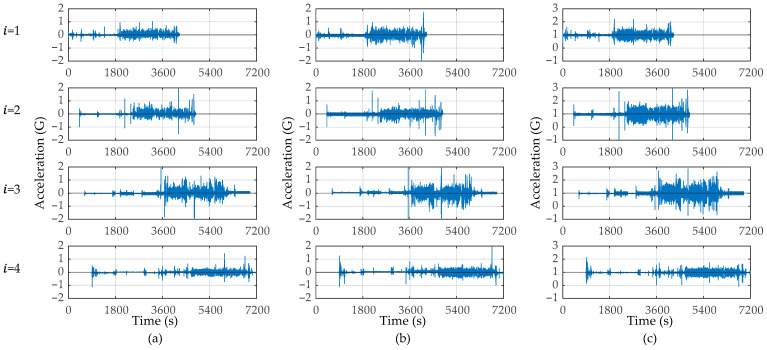
The measured vibration data: (**a**) acceleration of the travel direction, ${\ddot{x}}_{i}\left(t\right)=a_{i}\left(t\right)$; (**b**) acceleration of the lateral direction, ${\ddot{y}}_{i}\left(t\right)$; (**c**) acceleration of the vertical direction, ${\ddot{z}}_{i}\left(t\right)$.

**Figure 11 sensors-23-02004-f011:**
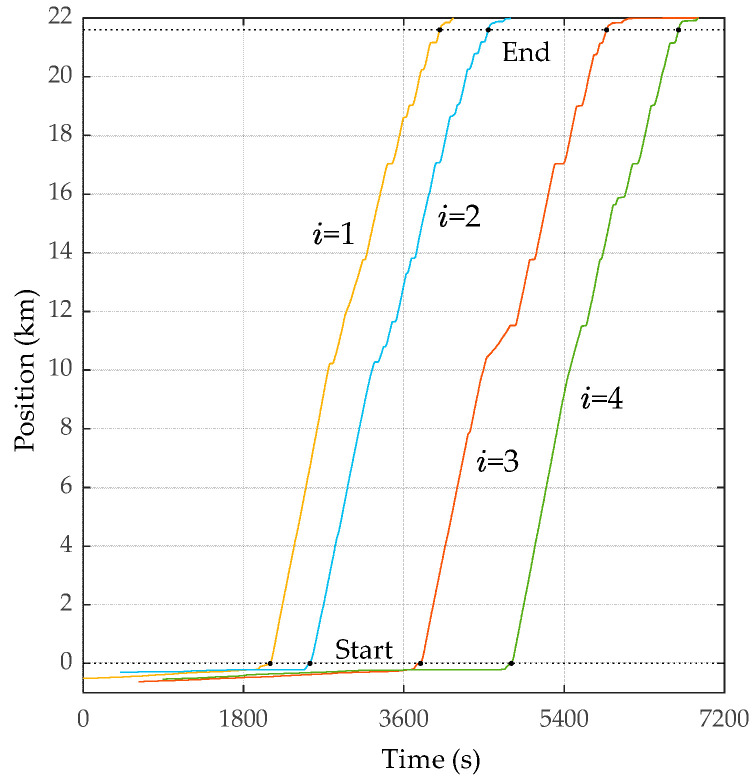
The x–t curve of each bus; the departure and arrival times of each bus are indicated by black dot markers. The slope of each curve represents the running speed. Note that $x_{i}\left(t\right)$ is the distance, not a coordinate.

**Figure 12 sensors-23-02004-f012:**
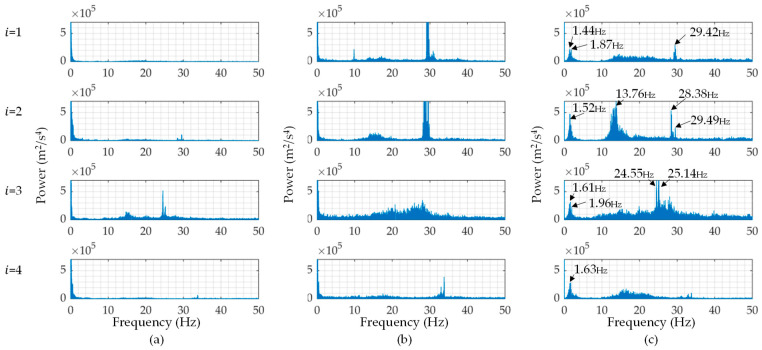
The Fourier’s power spectra of the measured vibration data. Each horizontal axis indicates the frequency, f ($=\frac{\omega }{2\pi }$): (**a**) travel direction, ${\hat{\ddot{x}}}_{i}\left(\omega \right)={\hat{a}}_{i}\left(\omega \right)$; (**b**) lateral direction, ${\hat{\ddot{y}}}_{i}\left(\omega \right)$; (**c**) vertical direction, ${\hat{\ddot{z}}}_{i}\left(\omega \right)$.

**Figure 13 sensors-23-02004-f013:**
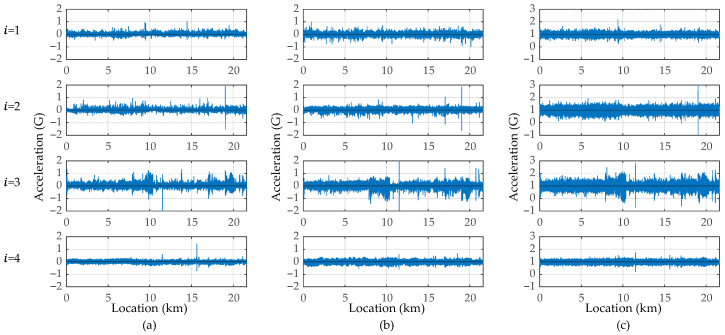
The spatially synchronized vehicle vibration data: (**a**) travel direction, ${\ddot{x}}_{i}\left(x\right)=a_{i}\left(x\right)$; (**b**) lateral direction, ${\ddot{y}}_{i}\left(x\right)$; (**c**) vertical direction, ${\ddot{z}}_{i}\left(x\right)$.

**Figure 14 sensors-23-02004-f014:**
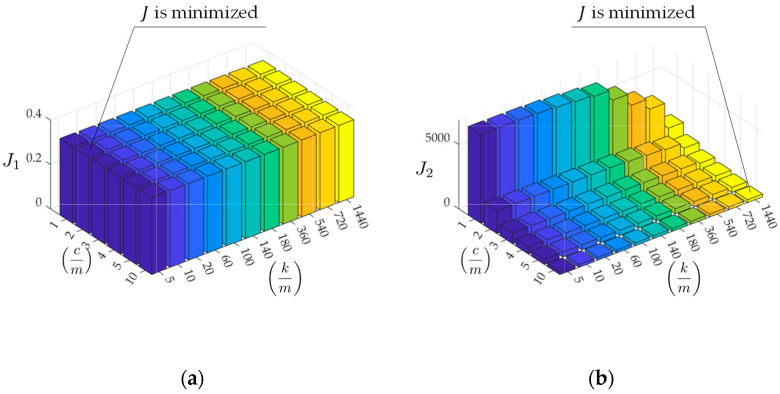
The values of objective functions: (**a**) based on Equation (9); (**b**) based on Equation (15).

**Figure 15 sensors-23-02004-f015:**
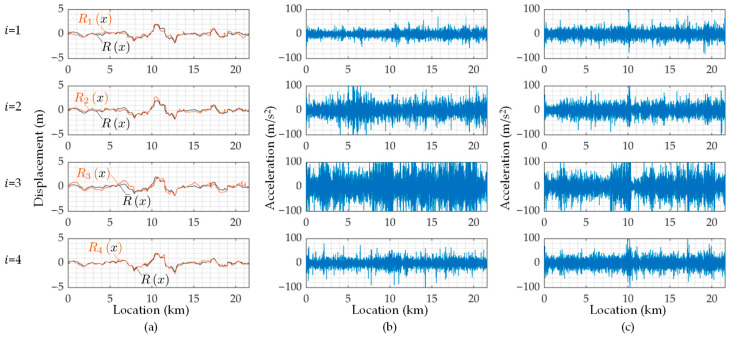
The estimated road profiles for $\left(\frac{c}{m}\right)$ = 2 and $\left(\frac{k}{m}\right)$ = 5: (**a**) the road profiles, $R_{i}\left(x\right)$, estimated from $\hat{r_{i}}\left(\omega \right)$; (**b**) the road profile accelerations, ${\ddot{R}}_{i}\left(x\right)$, estimated from $\hat{\ddot{r_{i}}}\left(\omega \right)$; (**c**) the road profile accelerations, ${\ddot{R}}_{i}\left(x\right)$, reproduced by $A\left(x\right)A{\left(x\right)}^{+}\ddot{r}\left(x\right)$.

**Figure 16 sensors-23-02004-f016:**
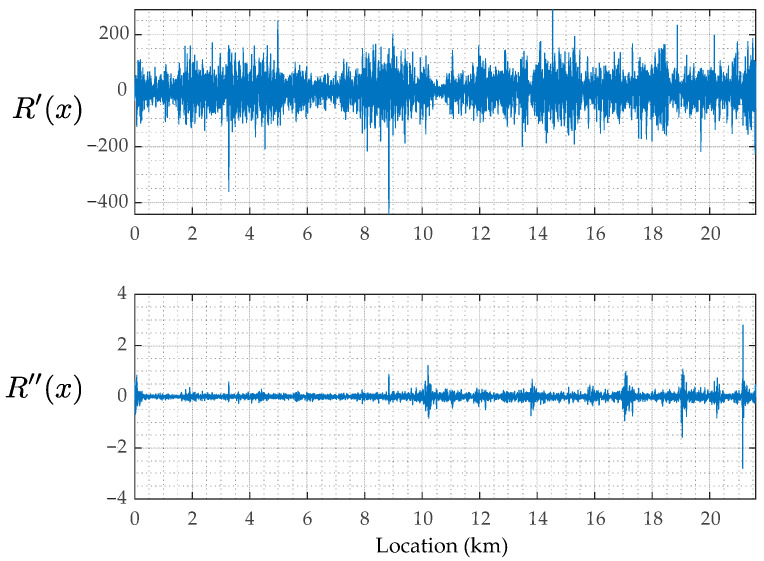
The estimated road unevenness components: $R^{\prime }\left(x\right)$ and $R^{\prime \prime }\left(x\right)$ for $\left(\frac{c}{m}\right)$ = 2 and $\left(\frac{k}{m}\right)$ = 5.

**Figure 17 sensors-23-02004-f017:**
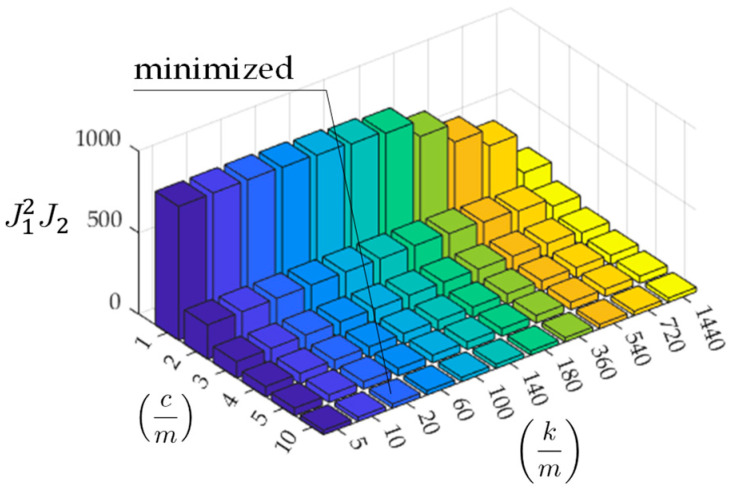
The product of objective functions: $J=J_{1}^{2}J_{2}$.

**Figure 18 sensors-23-02004-f018:**
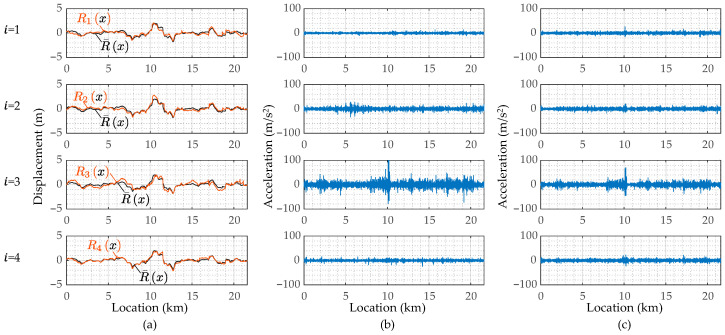
The estimated road profiles for $\left(\frac{c}{m}\right)$ = 10 and $\left(\frac{k}{m}\right)$ = 20: (**a**) the road profiles, $R_{i}\left(x\right)$, estimated from $\hat{r_{i}}\left(\omega \right)$; (**b**) the road profile accelerations, ${\ddot{R}}_{i}\left(x\right)$, estimated from $\hat{\ddot{r_{i}}}\left(\omega \right)$; (**c**) the road profile accelerations, ${\ddot{R}}_{i}\left(x\right)$, reproduced by $A\left(x\right)A{\left(x\right)}^{+}\ddot{r}\left(x\right)$.

**Figure 19 sensors-23-02004-f019:**
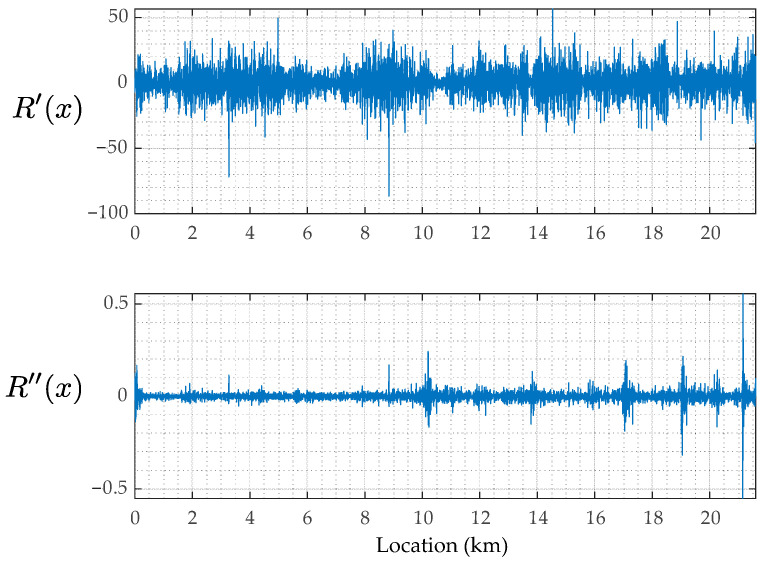
The estimated road roughness components: $R^{\prime }\left(x\right)$ and $R^{\prime \prime }\left(x\right)$ for $\left(\frac{c}{m}\right)$ = 10 and $\left(\frac{k}{m}\right)$ = 20.

**Table 1 sensors-23-02004-t001:** The parameters for identification.

Parameters											
$\frac{c}{m}$	1	2	3	4	5	10					
$\frac{k}{m}$	5	10	20	60	100	140	180	360	540	720	1440
Natural Frequency (Hz)	0.36	0.50	0.71	1.23	1.59	1.88	2.14	3.02	3.70	4.27	6.04

## Data Availability

Data is contained within the article.

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
