# Peer review of "Practical Application of Drive-By Monitoring Technology to Road Roughness Estimation Using Buses in Service"

_sensors, 2023, doi:10.3390/s23042004_

Round 1

Reviewer 1 Report

This study deals with the application of drive-by monitoring technology to road roughness estimation. The content is suitable to publish on Sensors. The manuscript was well written. The reviewer has some comments as follows:

1) For the generality, the word “in Rwanda” should be removed from the title.

2) There are many short paragraphs with only 2-3 sentences. They should be combined in the manuscript.

3) In section 2, the authors need to clarify the saved data, the data processing and the output results from the sensor.

4) In section 3.3, the authors need to clarify for the position xi(t), speed vi(t), and acceleration ai(t); which are measured directly, which are calculated from the other.

5) What is the sampling frequency for the data in Figure 10?

Reviewer 2 Report

1-    In the last line of the abstract, please quantify the research outputs.

2-    For lines 42 to 50, please provide citations.

3-    ‘This study does not measure the correct value of road unevenness” So what is its benefit?

4-    There was a misunderstanding between lines 106 and 107. Why can this be considered a contribution? This seems to have been taken into account in most previous methods.

5-    When you say travel vibrations, what do you mean?

6-    In a section before the conclusion, you should discuss the results and compare them to previous methods.

7-    Could you please provide more information about the field test, such as the type of road, IRI (if available), weather conditions, bus speed, etc.

Round 2

Reviewer 2 Report

No new comments!